# Association of Myocardial Infarction with CDKN2B Antisense RNA 1 (CDKN2B-AS1) rs1333049 Polymorphism in Slovenian Subjects with Type 2 Diabetes Mellitus

**DOI:** 10.3390/genes13030526

**Published:** 2022-03-16

**Authors:** Miha Tibaut, Franjo Naji, Daniel Petrovič

**Affiliations:** 1Department of Internal Medicine, General Hospital of Murska Sobota, Ul. dr. Vrbnjaka 6, 9000 Murska Sobota, Slovenia; 2Department for Cardiology and Angiology, Clinic for Internal Medicine, University Medical Centre Maribor, 2000 Maribor, Slovenia; franjo.naji@ukc-mb.si; 3Department of Internal Medicine, Faculty of Medicine, University of Maribor, 2000 Maribor, Slovenia; 4Institute of Histology and Embryology, Faculty of Medicine, University of Ljubljana, Korytkova 2, 1105 Ljubljana, Slovenia; 5International Centre for Cardiovascular Diseases MC Medicord.d., 6310 Izola, Slovenia

**Keywords:** Cyclin Dependent Kinase Inhibitor 2B Antisense RNA 1, CDKN2B-AS1, rs1333049, coronary artery disease, myocardial infarction, type 2 diabetes mellitus

## Abstract

Background: We examined the role of rs1333049 polymorphism of the CDKN2B Antisense RNA 1 (CDKN2B-AS1) on the prevalence of myocardial infarction (MI) in Slovenian subjects with type 2 diabetes mellitus (T2DM). Methods: A total of 1071 subjects with T2DM were enrolled in this retrospective cross-sectional case-control study. Of the subjects, 334 had a history of recent MI, and 737 subjects in the control group had no clinical signs of coronary artery disease (CAD). With logistic regression, we performed a genetic analysis of rs1333049 polymorphism in all subjects. Results: The C allele of rs1333049 polymorphism was statistically more frequent in MI subjects (*p* = 0.05). Subjects with CC genotype had a higher prevalence of MI than the control group in the co-dominant (AOR 1.50, CI 1.02–2.21, *p* = 0.04) and recessive (AOR 1.38, CI 1.09–1.89, *p* = 0.04) genetic model. Conclusions: According to our study, the C allele and CC genotype of rs1333049 polymorphism of CDKN2B-AS1 are possible markers of MI in T2DM subjects in the Slovenian population.

## 1. Introduction

Type 2 diabetes mellitus (T2DM) is a well-known metabolic disorder with considerable impact on human life as it affects around 463 million individuals worldwide [1]. Atherosclerosis in individuals with T2DM is accelerated [2], resulting in a greater prevalence of coronary heart disease (CAD) and, therefore, incidence of myocardial infarction (MI). Many modifiable and non-modifiable risk factors for the development of atherosclerosis in T2DM individuals are already identified. Still, there is an increasing tendency to seek out reliable markers with potential clinical impact [3,4].

Genetics plays a vital role in the development of CAD and MI. As CAD and MI are multifactorial diseases, the contribution of each identified loci to overall genetic susceptibility to its development is low. Still, the presence of multiple loci in an individual can alter its clinical course significantly. Up until now, at least 58 single nucleotide polymorphisms (SNP) were associated with CAD, contributing to around 13.3% of CAD heritability [5].

CDKN2B-AS1 gene is located within the CDKN2B-CDKN2A gene cluster at chromosome 9p21. It codes long non-coding RNA molecule which interacts with polycomb repressive complex-1 (PRC1) and -2 (PRC2), leading to epigenetic silencing of other genes in this cluster. Previous studies have shown that proteins from this gene cluster are richly expressed in atherosclerotic lesions and promote atherosclerosis through vascular remodeling, thrombogenesis, and plaque stability [6]. Moreover, genome-wide association studies (GWAS) identified polymorphisms from this locus associated with T2DM but have yet found no predictive value. Rs1333049, an SNP of CDKN2B-AS1 gene, has been connected to coronary artery disease and myocardial infarction in various populations by GWAS [7] and confirmed by single SNP studies [6]. There are several SNPs in this locus connected to cardiovascular diseases. Still, our study investigates rs1333049 since our research showed the most promise in connection to MI. To our knowledge, this is the first study to evaluate the correlation of rs1333049 polymorphism with MI in the Slovenian population with T2DM.

## 2. Materials and Methods

### 2.1. Subjects

This retrospective cross-sectional case-control study enrolled 1071 unrelated Caucasians with T2DM lasting no less than 10 years. Subjects were enrolled in General Hospital Murska Sobota. Participants were divided into two study groups: 334 subjects with MI and 737 subjects without a history of CAD, with no electrocardiographic signs of ischemic disease, and no ischemic changes during submaximal stress testing; clinically silent CAD was not an exclusion criterion. Participants were diagnosed with T2DM according to the current American Diabetes Association criteria [8]. Cases were diagnosed with history of MI according to the established universal standards [9]. Subjects with MI were included in the study 1 to 9 months after the acute event. Because of the lower incidence of MI in nondiabetic subjects, our study included only subjects with T2DM.

All participants enrolled in the study were Slovenians of Caucasian ethnicity. All subjects signed informed consent for participation in the study. Next, a detailed interview was performed, including active smoking status. Additionally, their blood was drawn for biochemical analysis and genotyping. Body mass index (BMI) was calculated as weight in kilograms divided by the height in meters square.

### 2.2. Ethical Statement

The national medical ethics committee (KomisijaRepublikeSlovenije za medicinskoetiko) (number 0120-372/2017) approved the study. The study abided by the Helsinki Declaration.

### 2.3. Biochemical Analyses

Using standard colorimetric assays on an automated biochemistry analyzer (Ektachem 250 Analyser, Eastman Kodak Company, Rochester, MN, USA) we determined total cholesterol, low-density lipoprotein cholesterol (LDL-c), high-density lipoprotein cholesterol (HDL-c), triglycerides (TG), and fasting glucose. We used the Friedewald formula to calculate the serum level of LDL-c. Hyperlipidemia was defined as total cholesterol higher than 5 mmol/L and/or TG higher than 2 mmol/L or as a condition treated with hypolipidemic medications.

High-performance liquid chromatography was used to estimate glycated hemoglobin (HbA1c) values. The average value of three recent HbA1c levels was used for each subject. Latex enhanced immunonephelometric assay was used to measure high sensitivity C-reactive protein (hsCRP).

### 2.4. Genotyping

Using a Qiagen isolation kit, genomic DNA was extracted from 100 μL of whole blood. The rs1333049 polymorphism of the CDKN2B-AS1 gene was genotyped by KBioscience Ltd. (LGC, Teddington, UK) using their own competitive allele-specific fluorescence-based PCR (KASPar) assay. For more details on the used method, refer to http://www.kbioscience.co.uk/, accessed on 2 May 2021.

### 2.5. Statistical Analysis

We expressed normally distributed continuous variables as means ± standard deviation and when asymmetrically distributed as median (interquartile range). The Kolmogorov–Smirnov test examined the normality of the continuous variables. An unpaired Student’s t-test was used to test normally distributed continuous variables and the Mann–Whitney U-test in asymmetrically distributed variables. We compared discrete variables with the Pearson χ^2^ test. Pearson χ^2^ test was similarly used to test whether the genotype distribution deviates from Hardy–Weinberg equilibrium. A significant relationship between two categorical variables was determined by the Fisher’s Exact test when cells with expected frequencies < 5 were identifiedin a contingency table.

Moreover, a stepwise multiple logistic regression was used for all variables that showed significant deviations in univariate analysis.

Statistical significance was considered with a *p*-value of ≤0.05. SPSS program version 19 (SPSS Inc., Chicago, IL, USA) was used for performing statistical analysis.

## 3. Results

Table 1 lists the clinical features and biochemical parameters of the cases and controls with T2DM. Cases (334 subjects with MI) had better-controlled hypertension, higher total cholesterol, LDL-c and triglycerides. Moreover, they had lower HDL-c, lower BMI, and waist circumference. Cases had a longer duration of T2DM diagnosis. Both groups were well-matched concerning age, gender, fasting glucose, HbA1c, hsCRP level, and concomitant history of CVI or TIA.

The genotype and allele frequencies of the rs1333049 polymorphism of the CDKN2B-AS1 gene are shown in Table 2. Subjects with MI and subjects without CAD, i.e., cases and controls, were in Hardy–Weinberg equilibrium regarding genotype distributions (cases: *p* = 0.24, controls: *p* = 0.74, Pearson χ^2^ test; respectively). CC genotype was more frequent in cases, but the difference did not reach statistical significance. On the other hand, the C allele was significantly more frequent in subjects with MI.

When binary logistic regression analysis for different genetic models was made, results showed an association of CC genotype with MI in T2DM patients in co-dominant and also in the recessive genetic model (Table 3). There was no association between cases and controls in the dominant genetic model. We adjusted odds ratios (AORs) estimates for the significant variables in the univariate analyses (Table 1): BMI, waist circumference, DBP, DM duration, triglycerides, total cholesterol, HDL-c, and LDL-c.

## 4. Discussion

Our study investigated the role of CDKN2B-AS1 gene polymorphism rs1333049 in Slovenian subjects with T2DM who recently experienced MI. We found an association between the C allele of rs1333049 polymorphism and MI. Furthermore, our study shows that in the co-dominant and recessive genetic model, the CC genotype is connected to MI compared to T2DM patients with no known CAD.

CDKN2B-AS1 gene is located on CDKN2A-CDKN2B gene cluster at the region of 9p21, well known for its connection with atherosclerotic diseases [10]. The product of the CDKN2B-AS1 gene is a long non-coding RNA molecule, which recruits PRC1 and PRC2 and polycomb complex protein EZH2. This, in turn, leads to activation of DNA methyltransferase (DNMT1), which increases epigenetic methylation of the DNA on CDKN2A and CDKN2B locus, thus inactivating them [11]. Transcriptions from CDKN2A and CDKN2B are part of a group of proteins called cyclin-dependent kinases (CDK) and cyclin-dependent kinase inhibitors (CKI). They are responsible for the regulation of the G1 cell cycle. CDKN2A protein p16 functions as an inhibitor of CDK4 and protein p14 as a stabilizer of p53, CDKN2B encodes CKI p15 that binds with CDK4 and CDK6 preventing subsequent activation of CDKs [11,12,13]. Inactivation of CDKN2A and CDKN2B locus thus reduces expression of p14, p15, and p16, which in turn stimulates proliferation of vascular smooth muscle cells (vSMC) [14], one of the critical factors in the process of atherosclerosis [15].

9p21 locus encompasses many SNPs connected to various atherosclerotic processes [16]. Rs1333049 polymorphism has been reported to be associated with hypertension development [17], cerebrovascular disease [18], Alzheimer’s disease [19], peripheral artery disease [20]. All of the correlations mentioned above can be understood through the mechanism of increased atherosclerosis. On the other hand, rs1333049 has been linked to breast [21], lung cancer, and chemotherapy toxicity [22], which can also be explained with CDKN2B-AS1 being indirectly responsible for the regulation of the cell cycle. 9p21 locus has been associated also with T2DM risk. The exact mechanism is not yet known but is thought to be a conjunction of effect on islet biology and other metabolic tissue.

Connection to CAD has also been established in rs1333049 polymorphism. Several meta-analyses [6,23] have confirmed this association with solid data for the European/Caucasian and Asian populations. The study from Dandona et al. [24] reported that subjects with rs1333049 risk genotype had three-vessel disease more frequently than the one-vessel disease; disease severity was directly associated with the number of risk alleles the subject was carrying. Wang et al. associated CC genotype with coronary plaque progression [25] and later to early-onset and increased severity of coronary artery disease [26] in a Chinese population; similar conclusions were made on Tunisian T2DM population [27], Asian-Indian [28], and Caucasian [29] populations.

On the other hand, studies looking for a connection of rs1333049 with MI gave divergent results. Several published studies [30,31,32,33,34,35] determined a positive relationship between CC genotype or C allele and MI. None of them was performed on the Slovenian population, but they used the “healthy” (without known CAD) population as a control group. Our study also found a positive connection with MI compared to the reference group of individuals with T2DM without known CAD. In-depth research of studies [29,36,37,38] that failed to find a positive connection with MI and rs1333049 polymorphism showed that in two of them, control groups consisted of subjects with established CAD [29,36]. Still, the other two compared cases to a healthy population and had a somewhat smaller sample. Since products of CDKN2B-AS1 promote the proliferation of vSMC, it would be logical to assume its connection with the development or progression of atherosclerosis and, therefore, CAD, but not with MI, whose main pathophysiological mechanisms are plaque rupture, erosion, erosion from calcium nodules, or intraplaque hemorrhage [15]. Each of mentioned mechanisms has a common prerequisite—rupture-prone plaque. One of the characteristics of rupture-prone plaques is the low vSMC number. More significant vSCM numbers tend to stabilize plaques with the production of connective tissue, therefore, forming a thick fibrous cap of atheroma. Promotion of vSMC’s migration and proliferation is similarly a mechanism of the plaque rupture healing process, restoring plaque integrity [15]. C allele in rs1333049 polymorphism of CDKN2B-AS1 reduces expression of p14, p15, and p16 with indirect stimulation of the proliferation of vSMC, consequently promoting atherosclerosis and CAD. Considering every mechanism mentioned above, rs1333049 should not alter the inclination towards MI but only facilitate the atherosclerotic process. On the contrary, a reasonable deduction would be that it would stabilize plaques by forming thick fibrous caps and lower the incidence of MI. One of the studies that addressed thisissue was from Patel et al. [39]. Patel et al. conducted a meta-analysis on 93,115 participants with established CAD and around 170,000 participants from other cohorts, including aCAD-free control group; the study showed an evident lack of association with MI in CAD patients but confirmed association with MI in CAD-free group. Our analysis yielded similar results, establishing aconnection with MI in CAD-free T2DM patients. However, to exclusively show a relationship with MI but not with its prerequisite—CAD, future study should be designed differently with a control group that would include CAD subjects without MI. On the other hand, most studies [30,31,32,33,34,35], including ours, looking into the connection of rs1333049 polymorphism with MI used control groups based on the absence of clinical signs of CAD and not on angiographically excluded CAD. Therefore, participants of our and such studies could also have clinically silent CAD, which affects results. Control group definition is, therefore, a significant limitation of our study.

## 5. Conclusions

To sum up, in our study, we showed an association of rs1333049 polymorphism with MI in T2DM subjects. C allele and CC genotype were also significantly more prevalent in the MI group in the co-dominant and recessive genetic model. Therefore, rs1333049 could potentially be used as a genetic marker to distinguish the subgroup of patients with T2DM with a more dire need for cardiologic evaluation. The conclusion is limited to the Slovenian population. Further investigations may be needed in other areas.

## Figures and Tables

**Table 1 genes-13-00526-t001:** Demographic and clinical characteristics of cases and controls in Slovenian subjects with T2DM.

	Case (*n* = 334) (Myocardial Infarction)	Control (*n* = 737) (Without CAD)	*p*-Value
Age (years)	64.3 ± 9.8	64.1 ± 9.1	0.75
BMI (kg/m^2^)	29.64 ± 4.14	30.69 ± 4.59	**<0.001**
Waist circumference (cm)	104.99 ± 11.42	107.86 ± 12.73	**0.02**
Male gender (%)	200 (59.9)	399 (54.1)	0.09
Systolic blood pressure (mm Hg)	148.1 ± 19.7	150.8 ± 19.7	0.06
Diastolic blood pressure (mm Hg)	82.11 ± 10.59	84.63 ± 11.52	**<0.001**
DM duration (years)	15 (10–23)	13 (10–18)	**<0.001**
Fasting plasma glucose (mmol/L)	8.87 ± 2.90	8.60 ± 2.54	0.25
Smoking prevalence (%)	43 (12.9)	66 (9.0)	**0.05**
Total cholesterol (mmol/L)	5.15 ± 1.45	4.64 ± 1.12	**<0.001**
HDL-c (mmol/L)	1.14 ± 0.30	1.24 ± 0.35	**<0.001**
LDL-c (mmol/L)	2.94 (2.22–3.76)	2.50 (2.02–3.10)	**<0.001**
Triglycerides (mmol/L)	1.90 (1.34–2.70)	1.60 (1.10–2.43)	**<0.001**
HbA1c (%)	7.88 ± 1.34	7.50 ± 1.9	0.64
hsCRP (mg/L)	2.40 (1.28–4.80)	2.40 (1.30–3.90)	0.15
CVI (%)	27 (8.1)	44 (6.0)	0.20
TIA (%)	17 (5.1)	21 (2.8)	0.07

Abbreviations: BMI, body mass index; HbA1c, glycated hemoglobin A1c; CVI, cerebrovascular insult; TIA, transitory ischemic attack; hsCRP, high-sensitivity C-reactive protein. Values in bold indicate statistical significance.

**Table 2 genes-13-00526-t002:** Genotype and allele frequencies distribution of rs1333049.

	Case (*n* = 334)	Control (*n* = 737)	*p*-Value
CC (MAF *)	96 (28.7%)	164 (22.3%)	
CG	156 (46.7%)	372 (50.5%)	0.07
GG	82 (24.6%)	201 (27.3%)	
C allele (%)	348 (52.1%)	700 (47.5%)	**0.05**
G allele (%)	320 (47.9%)	774 (52.5%)	
*p*-value (HWE)	0.24	0.74	

Abbreviations: HWE, Hardy–Weinberg equilibrium; * MAF, minor allele frequency. Values in bold indicate statistical significance.

**Table 3 genes-13-00526-t003:** Binary logistic regression analyses for the association between the rs1333049 of the CDKN2B-AS1 gene and MI in Slovenian subjects with T2DM.

Genetic Model	Case/Control	AOR (95% CI)	*p*-Value
Co-dominant			
CC vs. GG *	96/164 vs. 82/201	1.50 (1.02–2.21)	**0.04**
CG vs. GG *	156/372 vs. 82/201	1.14 (0.81–1.60)	0.46
Dominant			
[CC + CG] vs. GG *	252/536 vs. 82/201	1.06 (0.94–1.16)	0.44
Recessive			
CC vs. [CG + GG] *	96/164 vs. 238/573	1.38 (1.09–1.89)	**0.04**

* The reference; AOR, adjusted OR for BMI, waist circumference, DBP, total cholesterol, HDL-c and LDL-c, triglycerides, duration of DM in years; CI, confidence interval. The statistically significant result (*p*-value ≤ 0.05) is highlighted in bold.

## Data Availability

Data supporting reported results can be found locally in the Institute of Histology and Embryology, Faculty of Medicine, University of Ljubljana, Korytkova 2, 1105 Ljubljana, Slovenia.

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
