# Peer review of "Association of Myocardial Infarction with CDKN2B Antisense RNA 1 (CDKN2B-AS1) rs1333049 Polymorphism in Slovenian Subjects with Type 2 Diabetes Mellitus"

_genes, 2022, doi:10.3390/genes13030526_

Round 1

Reviewer 1 Report

In this paper, the authors investigated the associations between rs1333049 polymorphism and MI within the population those who have type 2 DM.  The analysis is based on a retrospective case-control study with 1071 subjects involved in Slovenian.  The study finds that C allele and CC genotype are significantly more prevalent in the MI group and rs1333049 can potentially be used as a biomarker in CAD evaluation. The topic is interesting, and the paper is well written with a clear logic.  

I have several minor comments to improve the paper.

  1. In the abstract, Line18, should be: ‘…cross-sectional case-control study’.
  2. In the abstract, Line20, suggest to say “We performed a genetic analysis of rs1333049 polymorphism in all subjects with logistic regression”.
  3. In Line101, you considered p-value < 0.05 as significant result. But in abstract, table 1 and table 2, you marked p=0.05 as significant result. I admit that this happens sometimes. Please address this paradox.
  4. Since you did not mention the missing data issue in the manuscript, I am curious how did you handle this issue? Please be explicit in the manuscript if necessary.
  5. In Table 1, please revise the column names to ‘Case’ and ‘Control’. You can also delete the item ‘Number’ and do like this, in the column names ‘Case (N=334)’ etc. Same for other tables.
  6. In Table2, p(HWE), p is for p-value, right? Please be consistent with the column name by using P-value.
  7. The study population is limited to in Slovenian. The scope of inference is expected to be contained in this country as well. That is, in Section 5, please consider adding the wording, like ‘The conclusion is limited to in Slovenia. Further investigations may be needed in other areas.’

Author Response

  • In the abstract, Line18, should be: ‘…cross-sectional case-control study’.
    • corrected
  • In the abstract, Line20, suggest to say “We performed a genetic analysis of rs1333049 polymorphism in all subjects with logistic regression”.
    • corrected
  • In Line101, you considered p-value < 0.05 as significant result. But in abstract, table 1 and table 2, you marked p=0.05 as significant result. I admit that this happens sometimes. Please address this paradox.
    • corrected
  • Since you did not mention the missing data issue in the manuscript, I am curious how did you handle this issue? Please be explicit in the manuscript if necessary.
    • Since all demographic and clinical characteristics of cases and controls were obtained during the initial inclusion examination and genetic information derived from whole blood taken at the time of examination, we had no missing data issue.

  • In Table 1, please revise the column names to ‘Case’ and ‘Control’. You can also delete the item ‘Number’ and do like this, in the column names ‘Case (N=334)’ etc. Same for other tables.
    • revised
  • In Table2, p(HWE), p is for p-value, right? Please be consistent with the column name by using P-value.
    • corrected
  • The study population is limited to in Slovenian. The scope of inference is expected to be contained in this country as well. That is, in Section 5, please consider adding the wording, like ‘The conclusion is limited to in Slovenia. Further investigations may be needed in other areas.’
    • added

Reviewer 2 Report

The study conducted by Tibaut and colleagues aims to evaluate the possible association of atherosclerosis-related rs1333049 gene polymorphism with prior MI prevalence in Slovenian individuals with type 2 DM. Based on a cross-sectional, case-control study design in 1071 subjects, the authors suggest the C allele and CC genotype of rs1333049 polymorphism of CDKN2B-AS1 as possible markers of MI in the above-mentioned population.

The topic is highly relevant, within the journal's scope, touching upon an important aspect of coronary artery disease. Overall the study design and results are interesting, although the specific SNP has been investigated in similar scenarios in the past.

Major comments

  • The lack of angiographic exclusion of CAD represents a limitation that the authors should acknowledge and discuss.
  • It would be interesting to analyze the study's outcome according to MI type (STEMI, NSTEMI, Unstable Angina).
  • Lines 190-197: the authors' point is unclear through the way it is presented. They are encouraged to provide additional clarity in this specific chapter.

Minor comments

  • Mild-moderate English changes required; please consult a native speaker or a word-processing software
  • Please provide information about the institution and the time period of patient recruitment in the methods section.
  • Line 77: Provide the reference for the Friedewald formula in the correct format.

Author Response

Major comments

  • The lack of angiographic exclusion of CAD represents a limitation that the authors should acknowledge and discuss.
    • acknowledgment and discussion added
  • It would be interesting to analyze the study's outcome according to MI type (STEMI, NSTEMI, Unstable Angina).
    • I agree but is not possible, since at the time of initial examination we did not collect information on MI subtypes.
  • Lines 190-197: the authors' point is unclear through the way it is presented. They are encouraged to provide additional clarity in this specific chapter.
    • Paragraph rephrased.

Minor comments

  • Mild-moderate English changes required; please consult a native speaker or a word-processing software
    • Grammarly premium used
  • Please provide information about the institution and the time period of patient recruitment in the methods section.
    • Provided
  • Line 77: Provide the reference for the Friedewald formula in the correct format.
    • Corrected

Round 2

Reviewer 2 Report

Thank you for the opportunity to review the revised version of the manuscript. The authors sufficiently addressed my comments. However, I think that the inability to assess the gene polymorphism in the different MI subtypes could also be added as a limitation. Other than that, I have no further comments.

Author Response

Thank you for your insight and positive review.